# Tissue-Specific Hormone Signalling and Defence Gene Induction in an In Vitro Assembly of the Rapeseed *Verticillium* Pathosystem

**DOI:** 10.3390/ijms241310489

**Published:** 2023-06-22

**Authors:** Fatema Binte Hafiz, Joerg Geistlinger, Abdullah Al Mamun, Ingo Schellenberg, Günter Neumann, Wilfried Rozhon

**Affiliations:** 1Department of Agriculture, Ecotrophology, and Landscape Development, Anhalt University of Applied Sciences, 06406 Bernburg, Germany; 2Institute of Crop Sciences, University of Hohenheim, 70593 Stuttgart, Germany

**Keywords:** priming, signal transduction, relative gene expression, beneficial microbes, *Verticillium longisporum*, phytohormones

## Abstract

Priming plants with beneficial microbes can establish rapid and robust resistance against numerous pathogens. Here, compelling evidence is provided that the treatment of rapeseed plants with *Trichoderma harzianum* OMG16 and *Bacillus velezensis* FZB42 induces defence activation against *Verticillium longisporum* infection. The relative expressions of the JA biosynthesis genes *LOX2* and *OPR3*, the ET biosynthesis genes *ACS2* and *ACO4* and the SA biosynthesis and signalling genes *ICS1* and *PR1* were analysed separately in leaf, stem and root tissues using qRT-PCR. To successfully colonize rapeseed roots, the *V. longisporum* strain 43 pathogen suppressed the biosynthesis of JA, ET and SA hormones in non-primed plants. Priming led to fast and strong systemic responses of JA, ET and SA biosynthesis and signalling gene expression in each leaf, stem and root tissue. Moreover, the quantification of plant hormones via UHPLC-MS analysis revealed a 1.7- and 2.6-fold increase in endogenous JA and SA in shoots of primed plants, respectively. In roots, endogenous JA and SA levels increased up to 3.9- and 2.3-fold in Vl43-infected primed plants compared to non-primed plants, respectively. Taken together, these data indicate that microbial priming stimulates rapeseed defence responses against *Verticillium* infection and presumably transduces defence signals from the root to the upper parts of the plant via phytohormone signalling.

## 1. Introduction

As sessile organisms, plants are exposed to diverse microbial communities in natural environments and have evolved a complex, multi-layered defence system to protect themselves against pathogen invasions in both local and distal tissues [1]. The root endophytic *Verticillium longisporum* is a soil-borne fungal pathogen from the Ascomycota phylum. *V. longisporum* is an amphidiploid hybrid that originated from four different ancestors via interspecific hybridization. It has a very restricted host range that mainly comprises Brassicaceae crops, including the economically important crop oil seed rape (*Brassica napus*) [2,3,4]. *V. longisporum* hyphae infect the rapeseed roots either through wounds or by the direct penetration of root hairs [5,6]. After entering the host system, hyphae colonize the root cortex both intercellularly and intracellularly, and invade the central cylinder [7]. Such colonization of the vascular system by *V. longisporum* hyphae results in the blockage of the vessels, which causes the obstruction of the sap stream in the xylem [8]. At the initial disease stage, dark unilateral stripes become visible on the stems of healthy plants in the late growing season due to cortical tissue necrosis caused by *V. longisporum* [9], and in later stages microsclerotia are formed in the stem cortex. After the decomposition of plant debris, microsclerotia enter the soil and reside there to start a new infection cycle upon the availability of suitable environmental conditions [10]. Disease symptoms develop with an increasing amount of pathogen-colonized rapeseed root and shoot tissues [11,12]. However, it is difficult to distinguish the disease symptoms from natural senescence due to premature crop ripening caused by *V. longisporum*. Since current disease control approaches are unable to provide appropriate protection, *V. longisporum* infestation management in rapeseed is very challenging [5]. Resistance breeding is impeded by the lack of effective resistance (R) genes against *V. longisporum*. However, recently, useful genetic resources capable of decreasing the susceptibility of brassicaceous plants to *V. longisporum* infection have been found. For example, the Enhancer of vascular Wilt Resistance 1 (EWR1) increased the resistance of *A. thaliana* against *V. longisporum* infestation significantly [13,14]. Nevertheless, no monogenic resistance has been yet detected against *V. longisporum*. In the C-genome of parental *B. oleracea* lines, three quantitative trait loci (QTLs) that notably correlate with *V. longisporum* resistance were identified. However, the understanding on actual mechanisms of this quantitative resistance is still insufficient [15]. Therefore, it is of high importance to develop alternative, cost-effective and efficient strategies to control *V. longisporum* infection in rapeseed.

In response to pathogen attacks, plants have developed a number of inducible defence mechanisms for protection [16,17], including induced systemic resistance (ISR), triggered by beneficial fungi, e.g., *Trichoderma harzianum* [18,19,20], and rhizobacteria, e.g., *Bacillus velezensis* [21,22], and systemic acquired resistance (SAR), triggered by pathogenic fungi, e.g., *V. longisporum* [23], bacteria or insects [24,25]. Although ISR and SAR share several signalling components, these two phenomena are different from each other based on elicitors and the regulatory pathways. In systemic tissues, the initiation of ISR requires the activation of jasmonic acid (JA) and ethylene (ET)-regulated pathways [26,27]. In contrast, SAR is characterized by increased salicylic acid (SA) levels, which subsequently enhance the expression of pathogenesis-related (PR)-protein coding genes through the activation of the redox-regulated protein Non-Expressor of PR genes 1 (NPR1) [28,29,30,31,32,33,34,35]. In contrast to SAR, in ISR, plant-beneficial microbes stimulate the defence system of the whole plant body for fast and strong responses against pathogen attack instead of directly activating local defence responses. This phenomenon is called defence priming [28,32,36,37,38,39]. To enter the plant cells and subsequently establish the symbiotic relationship, beneficial rhizosphere-associated microbes suppress the host defence system by modulating plant hormone signalling pathways [30]. Induced defence responses of plants against pathogenic infection rely on the inception of ET/JA and SA phytohormone-mediated signalling pathways [40,41]. In nature, the crosstalk of phytohormones is the backbone of plants’ defence responses against biotrophic and necrotrophic pathogens [42,43]. A detailed analysis of *A. thaliana* mutants impaired in SA, JA and ET signalling, singly or in combination, revealed that defence signalling is not mediated individually, but rather orchestrated in a complex network in which SA, ET and JA hormones extensively crosstalk [44,45,46,47,48]. Studies also reported both synergistic and antagonistic functions of the SA-, ET- and JA-dependent defence signalling pathways [32,49,50,51,52].

In this study, we investigated the role of *T. harzianum* OMG16 and *B. velezensis* FZB42 priming in rapeseed growth responses and defence activation against *V. longisporum* 43 (Vl43) infection. Therefore, the relative expressions of ET, JA and SA phytohormone biosynthesis, as well as signalling genes, were analysed separately in root, stem and leaf tissues of Vl43-infected primed and non-primed rapeseed plants by applying qRT-PCR. Moreover, the contents of the endogenous phytohormones JA and SA were quantified in both shoot and root tissues after Vl43 infection in both primed and non-primed rapeseed using UHPLC-MS analysis.

## 2. Results

### 2.1. Analysis of V. longisporum Growth in Rapeseed

To determine the substantial virulent interaction between *Verticillium* and rapeseed, roots of two-week-old rapeseed plants were infected with the spores of the pathogenic fungus Vl43 strain. A microscopic analysis of royal blue ink-stained leaves 8 days after root infection with Vl43 revealed a net of hyphae colonizing the rapeseed leaves (Figure 1). This confirmed that spores germinated, and hyphae were able to penetrate rapeseed roots, colonize the xylem and move into the leaves via the vascular system.

### 2.2. Effects of Priming on Rapeseed Growth Performance

To determine the effects of microbial priming on plant growth and development, the fresh weights of root and shoot tissues of primed and non-primed rapeseed seedlings were measured. In Vl43-infected primed plants, the fresh weight of roots was increased 2 days after infection (dai) and 5 dai (*p* < 0.0001 each) compared to the untreated controls. In contrast, the root weight was decreased on 3 dai (*p* < 0.01) in the Vl43-infected non-primed plants compared to the untreated controls. In Vl43-infected primed plants, the root weight was highly enhanced on 2 dai (*p* < 0.0001), 3 dai (*p* < 0.001) and 5 dai (*p* < 0.0001) compared to the non-primed plants (Figure 2A).

In addition, the shoot weight of Vl43-infected non-primed rapeseed plants slightly (*p* < 0.05) increased as late as on 5 dai compared to the untreated control plants. Microbial priming also enhanced the shoot weight 5 days after Vl43 infection (*p* < 0.0001), compared to the untreated controls. In addition, the shoot fresh weight increased in Vl43-infected primed plants on 5 dai (*p* < 0.01) compared to the non-primed plants (Figure 2B).

### 2.3. Effects of Priming on V. longisporum 43 Root Infection

The effects of OMG16 plus FZB42 priming on reducing the Vl43 infection rate in rapeseed roots were analysed using qPCR (absolute quantification). The average amount of Vl43 DNA was substantially lower in primed plants than in non-primed plants (Table 1). Vl43 was detectable in Vl43-infected non-primed roots as early as on 1 dai, where an amount of 40.8 ng Vl43 DNA in 15 ng total root DNA was observed. In the following days, the amount continued to increase until 5 dai, the end point of this experiment, where an amount as high as 271.7 ng Vl43 DNA in 15 ng total root DNA was reached. In sharp contrast, the average amount of Vl43 DNA in roots of primed plants was significantly lower, and reached only 15.2 ng Vl43 DNA in 15 ng total root DNA, even at 5 dai. These data demonstrate the high efficiency of priming for restricting infection by *V. longisporum*, and are in line with our previous findings in Hafiz et al. (2022) [53].

### 2.4. Enhanced Rapeseed Defence upon Priming

#### 2.4.1. The Expression of Defence-Related Rapeseed Genes in Roots

To investigate the local defence responses of the plants, the relative transcript abundances of the JA biosynthesis genes *Lipoxygenase 2* (*LOX2*) and *12-Oxophytodienoic acid reductase 3* (*OPR3*), the ET biosynthesis genes *1-Aminocyclopropane-1-carboxylate* (*ACC*) *synthase 2* (*ACS2*) and *ACC oxidase 4* (*ACO4*), the SA biosynthesis gene *Isochorismate synthase 1* (*ICS1*) and the SA-specific marker *pathogenesis-related gene 1* (*PR1*) were analysed in root tissue using qRT-PCR. Prior to the pathogen infection, there was no significant induction of these defence genes in roots of OMG16- and FZB42-primed plants compared to the untreated controls (Appendix A).

The expression of both *LOX2* and *OPR3*, involved in JA biosynthesis, was rapidly downregulated in roots of Vl43-infected non-primed plants during the whole experimental period in comparison with non-infected controls. However, microbial priming enhanced *LOX2* and *OPR3* expression as early as on 1 dai until the end point of this experiment, compared to the Vl43-infected non-primed plants (Figure 3A,B).

The analysis of ET biosynthesis gene expression in the roots of Vl43-infected non-primed plants revealed that the mRNA abundances of *ACS2* and *ACO4* were downregulated compared to the untreated controls. However, this effect was completely reverted by priming the roots with the selected OMG16 and FZB42 inoculants prior to Vl43 infection (Figure 3C,D).

Similarly, the expressions of SA biosynthesis *ICS1* and SA response *PR1* were suppressed in roots of Vl43-infected non-primed plants compared to the untreated control plants. However, both *ICS1* and *PR1* expressions were higher in Vl43-infected primed than in the Vl43-infected non-primed plants (Figure 3E,F).

#### 2.4.2. The Expression of Defence-Related Genes in Stems

In addition, to investigate the systemic defence responses, the relative transcript abundances of the previously studied genes were analysed in stem tissue using qRT-PCR. Prior to infection, OMG16 and FZB42 priming significantly increased the relative expression of *LOX2*, *OPR3* and *ICS1* in stems compared to the untreated controls (Appendix A).

Unlike in roots, the expression of the JA biosynthesis gene *LOX2* was highly enhanced in stems of Vl43-infected non-primed plants as early as 1 dai until 5 dai, the end point of the experiment, compared to the untreated controls. In contrast, the *OPR3* expression increased only on 2 dai. Moreover, both *LOX2* and *OPR3* expression increased in stems of Vl43-infected primed plants compared to Vl43-infected non-primed plants (Figure 4A,B).

In the stems of non-primed plants, the expression of the ET biosynthesis genes *ACS2* and *ACO4* behaved differently in primed and non-primed Vl43-infected plants. In primed Vl43-infected plants, the expression of *ACS2* remained, at most time points, at relatively low values, similar to control plants, while the transcript level was clearly increased in non-primed, Vl43-infected plants (Figure 4C). On the other hand, the expression of *ACO4* was massively increased in primed, Vl43-infected plants compared to the untreated controls, while in Vl43-infected non-primed plants it was only slightly enhanced or remained even at a level comparable to control plants (Figure 4D).

In both primed and non-primed plants, the expression of the SA biosynthesis gene *ICS1* only increased on 3 dai (*p* < 0.01 each) after infection. Interestingly, the expression of *ICS1* was higher on 2 dai (*p* < 0.01) in Vl43-infected non-primed plants compared to the primed plants (Figure 4E).

The expression of *PR1* in stems of Vl43-infected non-primed plants was already elevated on 1 dai and remained upregulated until 5 dai compared to the untreated controls. However, microbial priming enhanced the *PR1* expression on 1 dai and 2 dai in Vl43-infected primed plants compared to the non-primed ones. In contrast, on 5 dai, *PR1* was higher (*p* < 0.05) in Vl43-infected non-primed plants compared to the primed plants (Figure 4F).

#### 2.4.3. The Expression of Defence-Related Genes in Leaves

To assess the systemic defence responses of rapeseed, the relative expressions of the previously analysed genes were measured in leaf tissue using qRT-PCR. Upon priming, the relative expression of *OPR3*, *ACO4* and *PR1* significantly enhanced in leaves of OMG16- and FZB42-primed plants prior to Vl43 infection, compared to the untreated controls (Appendix A).

Vl43 infection delayed the induction of *LOX2* and *OPR3* expression in leaves compared to the untreated controls. Priming with OMG16 and FZB42 led to a higher transcript abundance of these two JA biosynthesis genes in leaves after Vl43 root infection than the Vl43-infected non-primed plants (Figure 5A,B).

The late induction of *ACS2* was observed in Vl43-infected non-primed plants compared to the untreated controls (Figure 5C). In contrast, Vl43 infection enhanced *ACO4* expression in non-primed plants during the whole experimental period (Figure 5D). Priming increased the expression of both *ACS2* and *ACO4* after Vl43 infection more rapidly than in Vl43-infected non-primed plants.

The expression of *ICS1* and *PR1* was only enhanced as late as on 5 dai in Vl43-infected non-primed plants compared to the untreated controls. For both investigated genes, this effect was accelerated by priming the roots with the beneficial microorganisms, OMG16 and FZB42 (Figure 5E,F).

### 2.5. Determination of Stress-Related Phytohormonal Homeostasis

To analyse the variation in endogenous phytohormone content after priming, the amounts of plant hormones present in rapeseed seedlings were determined in shoots (both in leaves and stems) and roots. The 14-day-old *B. napus* seedlings were primed with OMG16 plus FZB42. Two days after priming, roots were infected with the pathogenic Vl43 fungal strain. The amounts of endogenous phytohormones JA and SA were measured with UHPLC-MS.

#### 2.5.1. Phytohormone Contents in Root Tissue

In root tissue, the JA content was only significantly lower on 5 dai (*p* < 0.05) in Vl43 infected primed plants compared to the untreated controls. In contrast, in Vl43-infected non-primed plants, JA production was reduced on 1 dai, 2 dai, 3 dai and 5 dai (*p* < 0.05 each) compared to the untreated controls. Nevertheless, the JA level highly increased in the roots of Vl43-infected primed plants on 1 dai (*p* < 0.001) and 2 dai (*p* < 0.0001) compared to the non-primed ones (Figure 6A).

The content of SA increased on 2 dai (*p* < 0.001), 3 dai (*p* < 0.01) and 5 dai (*p* < 0.05) in Vl43-infected primed plants compared to the untreated controls. In contrast, there was no induction of SA in Vl43-infected non-primed plants compared to the untreated control plants. The level of endogenous SA was enhanced in roots of Vl43-infected primed plants on 2 dai (*p* < 0.01), 3 dai (*p* < 0.05) and 5 dai (*p* < 0.05) compared to the non-primed controls (Figure 6B).

#### 2.5.2. Phytohormone Contents in Shoot Tissues

The content of JA increased as late as on 5 dai (*p* < 0.05 each) in shoots of both Vl43-infected primed and non-primed plants compared to the untreated controls. The JA content also increased in Vl43-infected primed plants on 2 dai (*p* < 0.05) compared to the Vl43-infected non-primed plants (Figure 6C).

The endogenous content of SA was enhanced only on 1 dai (*p* < 0.05) in shoots of Vl43-infected primed plants compared to the untreated controls (Figure 6D). There was no induction of the SA level in Vl43-infected primed plants compared to the non-primed ones.

## 3. Discussion

### 3.1. Microscopic Observation of V. longisporum Growth in Rapeseed Leaves

The micrograph presented in Figure 1 shows the hyphal net of Vl43 endophytically colonizing the rapeseed leaves 8 days after root infection. After entering the rapeseed roots, Vl43 hyphae spread to the leaves through the vascular system. This hyphal migration process was previously described by Eynck et al. (2007) and Zheng et al., 2019 [7,54]. This observation also confirmed that the Vl43 spores used were vital and virulent.

### 3.2. Priming Effects on Plant Growth

The application of combined OMG16 plus FZB42 inocula to the rapeseed roots prior to Vl43 infection improved plant growth. Data indicated that the fresh weight of rapeseed root increased up to 1.9-fold in primed plants compared to the non-primed seedlings after Vl43 infection (Figure 2A), while the shoot fresh weight of primed plants enhanced up to 1.3-fold compared to the non-primed rapeseed followed by Vl43 infection (Figure 2B). In line with this result, previous investigations demonstrated that *V. longisporum* infection reduced the biomass of rapeseed, and the root biomass was more severely decreased than the shoot. Interestingly, the root colonization by OMG16 with five different *Bacillus* strains improved the growth and morphology of maize roots [55]. Similar effects were also reported for a *Bacillus* consortium, where PGPR treatment induced plant growth and development, which in turn increased the biomass and phenotype of maize leaves [56].

### 3.3. Priming Effects on Reducing Root Infection by Vl43

The priming of rapeseed roots with OMG16 plus FZB42 successfully reduced Vl43 root infection. Approximately 10 to 100 times less infection as measured by presence of pathogen DNA was recorded in the primed roots compared to the non-primed (Table 1). The underlying mechanisms of such disease resistance might include the physical barrier formed by root endophytic OMG16 and FZB42 microbes, restricting the entry and growth of Vl43 hyphae into the rapeseed roots. Here, it is worth mentioning that our previous study demonstrated OMG16-formed endophytic hyphae inside rapeseed root hairs to develop distinct intracellular structures [53]. Other reasons for Vl43 growth suppression might include competition for space and nutrition [57], the production of hydrolytic enzymes by the *Trichoderma* strain to break down the cell wall of the pathogen and the release of secondary antimicrobial metabolites [58,59].

### 3.4. Priming Effects on Rapeseed-Enhanced Defence Activation after Vl43 Pathogen Challenge

Our previous study demonstrated that OMG16 and FZB42 priming reduced Vl43 growth up to 100-fold in rapeseed root tissue [53]. As we show here, priming also enhanced the plants’ systemic defence responses against Vl43 infection (Figure 3, Figure 4 and Figure 5). Priming the rapeseed roots significantly enhanced the JA and ET biosynthesis, as well as the SA signalling pathways in leaves, by inducing the corresponding biosynthesis and signalling genes, *OPR3*, *LOX2* and *PR1* (Appendix A). This observation corroborates previous studies reporting that *T. virens* and *T. atroviride* activated the JA/ET pathway by inducing the expression of *LOX-1* in *A. thaliana*. The same fungi also induced the expression of *PR1*, thereby activating the SAR pathway [60,61]. In addition, *T. asperellum* DQ-1 enhanced the expression of *PR2* and *LOX1* in both *Botrytis cinerea* challenged and non-challenged inoculation treatments, and increased resistance against grey mould disease in tomato [62].

Similarly, in stems, priming also showed fast and strong induction of the JA biosynthesis pathway and relatively slower and weaker induction of SA biosynthesis (Appendix A). There was no significant induction of JA, ET and SA biosynthesis or signalling in roots (Appendix A). These data indicate that beneficiary fungal and bacterial strains can prime the whole rapeseed plant body and transduce the ISR defence signal into the upper part of plants, presumably via the phytohormones JA and ET or through some as-yet-unknown mobile messenger. Subsequently, when the rapeseed roots were infected with Vl43 spores 2 days after priming, such molecular memory of immunization enhanced rapeseed defence against the pathogen challenge. Similar effects were previously observed after the combined application of *T. harzianum* Tr6 and *Pseudomonas* sp. Ps14, which significantly increased the resistance level in cucumber against *Fusarium oxysporum* infection through the elevated expression of a set of defence-related genes [63].

The analysis of gene expression using qRT-PCR revealed a fast and strong induction of JA, ET and SA biosynthesis and signalling genes separately in root, stem and leaf tissues of the OMG16 plus FZB42 primed plants. Moreover, non-primed Vl43 infection also induced rapeseed defence through the SAR pathway by enhancing the expression of JA, ET and SA biosynthesis and signalling genes. Such *Verticillium*-mediated defence activation was relatively weaker and slower.

#### 3.4.1. Priming Effects on Roots

qRT-PCR analysis showed that the expression of *LOX2* and *OPR3* was downregulated in the roots of Vl43-infected non-primed plants (Figure 3A,B). This indicates that for the successful colonization of rapeseed roots, Vl43 suppressed the JA biosynthesis pathway shortly after infection (1 dai) and kept it attenuated at least until 5 dai, the end point of this experiment. A previous study demonstrated that Vl43 requires the functional activity of JA receptor, coronatine insensitive 1 (COI1), in the roots of *Arabidopsis thaliana* to efficiently colonize its roots and elicit disease symptoms in the shoots [64]. In contrast, rapeseed plants primed with OMG16 and FZB42 were able to successfully overcome the suppression of JA biosynthesis in the roots after Vl43 infection. The upregulation of both *LOX2* and *OPR3* in the primed plants was identified as fast as 1 dai, and remained activated until 5 dai compared to the non-primed plants. Nevertheless, the expression of *OPR3* still remained downregulated in Vl43-infected primed plants compared to the untreated controls. However, *OPR3* expression was remarkably higher in the Vl43-infected primed plants than in the non-primed plants.

In addition, endogenous JA contents in rapeseed roots were measured using UHPLC-MS. The JA level was approximately 10-fold less in the roots of Vl43-infected non-primed plants than in the untreated controls (Figure 6A). Vl43 repressed JA production in rapeseed roots as early as 1 dai, and this continued until 5 dai. This observation again demonstrated that Vl43 infection impaired JA biosynthesis during the early root colonization process. In Vl43-infected primed plants, the JA content was 3.9-fold higher within 1 dai and 2 dai than in Vl43-infected non-primed plants. This points to the evidence that priming with OMG16 and FZB42 enabled the plants to overcome the suppression of JA biosynthesis caused by Vl43.

Vl43 infection also inhibited the ET biosynthesis pathway at the site of infection by downregulating the expression of *ACS2* and *ACO4* in non-primed plants (Figure 3C,D). Compared to the untreated controls, *ACS2* expression was suppressed as late as 5 dai, while the downregulation of *ACO4* was detected as early as on 2 dai and remained suppressed until 5 dai. The phytohormone ET was reported earlier to be necessary for *V. longisporum* to efficiently colonize plant roots. The *A. thaliana* mutants, *ein2-1*, *ein4-1* and *ein6-1*, impaired in the ET signalling pathway, displayed an enhanced susceptibility to *V. longisporum* compared to wild-type plants [65,66]. Rapeseed roots previously treated with OMG16 and FZB42 suspensions showed fast and strong induction of the ET biosynthesis pathway by enhancing the expression of *ACS2* and *ACO4* in root tissues after Vl43 infection. In addition, the expression of the ET biosynthesis genes *ACS2* and *ACO4* also reacted faster and more strongly to Vl43 infection in primed compared to the non-primed plants. This points to the fact that priming with beneficial microbes enabled rapeseed roots to overcome the adverse effects of ET biosynthesis caused by Vl43 infection and enhanced the generation of ET.

The expression of *ICS1* and *PR1* was drastically reduced after Vl43 infection in the non-primed roots, indicating that the Vl43 pathogen suppressed both SA biosynthesis and signalling pathways to successfully colonize the rapeseed roots (Figure 3E,F). In accordance with this result, no induction of *PR1* was found in the Vl43-infected rapeseed roots at 3 dai [67]. In a COI1-dependent manner, *V. longisporum* suppressed the SA-dependent defence in rapeseed by generating JA-isoleucine-mimic coronatine [64]. In our study, we showed that primed plants were able to quickly enhance the expression of both *ICS1* and *PR1*, thus mitigating the downregulation of SA biosynthesis and signalling caused by Vl43 infection. Moreover, the endogenous SA content was also enhanced in Vl43-infected primed roots compared to both Vl43-infected non-primed plants and untreated control plants (Figure 6B). The synthesis of SA in the roots of primed rapeseed increased to 2.3-fold on 2 dai and remained induced until 5 dai. Ratzinger et al. (2009) reported that SA is involved in the susceptibility of *B. napus* to *V. longisporum* infection [68]. Priming may reduce the susceptibility by increasing SA accumulation in rapeseed roots.

Taken together, these results suggest that Vl43 hijacks the defence system of rapeseed by inhibiting the biosynthesis of JA, ET and SA phytohormones, and successfully enters into the roots to establish a long-lasting compatible interaction with the host. Rapeseed roots primed with OMG16 and FZB42 microbes revealed a rapid reprogramming of gene expression to activate the biosynthetic pathways of the hormones JA, ET and SA, and restricted the entry of the Vl43 pathogen into the root cells. In a similar study, the inoculation of watermelon roots with the biocontrol agent F1-35 induced the JA and ET pathways by elevating the expression of *LOX1*, *AOS* and *JAR1* in local root tissue. Inoculated plants showed improved resistance against *Fusarium oxysporum* f. sp. *Niveum* [69].

#### 3.4.2. Priming Effects on Stems

After penetrating the root, the pathogen Vl43 systemically spread through the xylem and generated microsclerotia in the stem of oil seed rape to develop “*Verticillium* stem striping” disease [5,7]. Gene expression analysis with qRT-PCR revealed a fast and strong induction of *LOX2* and *OPR3* in the stem tissue (Figure 4A,B). The expression of these two JA biosynthesis genes indicated the enhanced production of JA in both Vl43-infected primed and non-primed plants. Notably, the expressions of *LOX2* and *OPR3* were higher in the stems of Vl43-infected primed plants compared to the non-primed controls.

Similarly, the expressions of ET biosynthesis genes *ACS2* and *ACO4* were highly enhanced in stems of both Vl43-infected primed and non-primed plants (Figure 4C,D). The *ACO4* expression was higher in Vl43-infected primed than non-primed plants. Interestingly, after the Vl43 challenge, the *ACS2* expression increased in the non-primed plants compared to the primed ones.

In Vl43-infected stems, SA biosynthesis and signalling increased both in primed and non-primed plants. *PR1* expression was higher in the primed stems for the first 2 days after infection, while the expression increased on 5 dai in the non-primed plants (Figure 4F). This indicates that SA signalling increased in the stem tissue with the gradual progress of Vl43 infection. In line with this result, a previous study documented an elevated level of SA and its metabolite, salicylic acid glucoside (SAG), in the xylem sap of rapeseed shoots after *V. longisporum* infection. A strong correlation was also reported between the content of *V. longisporum* DNA in hypocotyls and SAG levels in the shoot [68].

#### 3.4.3. Priming Effects on Leaves

The analysis of gene expression with qRT-PCR revealed the faster and stronger induction of *LOX2* and *OPR3* in leaves of Vl43-infected primed plants compared to non-primed plants (Figure 5A,B). The expression of these two biosynthesis genes in leaves indicated the activation of the JA biosynthesis pathway in both Vl43-infected primed and non-primed plants. Moreover, endogenous JA levels were up to 1.7-fold higher in the shoots of Vl43-infected primed than in non-primed plants (Figure 6C). Therefore, these data indicated that the presence of OMG16 and FZB42 in rapeseed roots induced the JA biosynthesis pathway in leaves.

The expression of ET biosynthesis genes *ACS2* and *ACO4* was highly enhanced in the leaves of primed plants after Vl43 infection (Figure 5C,D). Therefore, these results demonstrated that OMG16 and FZB42 priming induced the ET biosynthesis pathway in rapeseed leaves to increase the resistance against Vl43 infection.

In addition, the expression of the SA biosynthesis gene *ICS1* and the SA signalling gene *PR1* was remarkably enhanced in leaves of Vl43-infected primed plants (Figure 5E,F). Moreover, the level of endogenous SA was elevated up to 2.6-fold in shoots of Vl43-infected primed plants shortly after infection (1 dai) (Figure 6D), which indicates that both SA biosynthesis and signalling pathways were induced in leaves after priming the rapeseed roots. In a previous study, seed-coating by *Trichoderma atroviride* SG3403 and *Bacillus subtilis* 22 co-culture enhanced the expression of defence genes, including *PR1* and *ACS*, which promoted tolerance against *F. graminearum* infection in wheat leaves [70].

Similarly, the pre-inoculation of peanut with the root endophytic fungus *Phomopsis liquidambaris* B3 triggered SA signalling by enhancing the expression of *PRs* and increasing the SA level, which in turn reduced *F. oxysporum* root colonization and disease severity both locally and systemically. However, such resistance was absent in the untreated roots [71].

Taken together, our data indicate that the priming of rapeseed roots with the beneficial microbes OMG16 and FZB42 can lead to rapid and strong induction of plant defence hormone biosynthesis and signalling in roots, stems and leaves. Moreover, a root-borne mobile signal transduces from roots via stems to leaves, mediated by the phytohormones JA, ET and SA, to enhance the defence system in response to Vl43 infection. Similarly, *B. amyloliquefaciens* ssp. *plantarum* UCMB5113 rhizobacterium enhanced plant defence priming and conferred resistance to airborne pathogens without any direct interaction [72]. In a similar study, rhizobacterium *P. alvei* K165 triggered ISR to protect *A. thaliana* from *V. dahliae* infestation. K165 induced the SA-dependent defence pathway by increasing the *PR1* expression in leaves [73]. On the other hand, the transcriptome analysis of *Trichoderma atroviride* T11 showed increased expression levels of genes involved in catalytic, hydrolytic and transporter activities, and secondary metabolism in response to *V. dahliae* infection [74]. Therefore, the orchestrated transcriptional reprogramming of JA, ET and SA pathways upon priming induced rapeseed defence and restricted *V. longisporum* proliferation.

## 4. Materials and Methods

### 4.1. Plant Materials and Growth Conditions

Oilseed rape (*B. napus*) seedlings, genotype AgP4 (seeds were provided by the NPZ Innovation GmbH, Hohenlieth, Germany), were used for all the experiments in this study. Details of the cultivar were discussed in Hafiz et al. (2022) [53]. For surface sterilization, *B. napus* seeds were incubated in 3% sodium hypochlorite solution (Carl Roth, Karlsruhe, Germany) for 15 min with gentle shaking followed by rigorous washing with sterile, deionised water. Surface-sterilized seeds were germinated on soaked filter papers (grade MN 619 eh; Macherey-Nagel, Düren, Germany) placed on glass Petri dishes (150 × 30 mm), placed in a growth chamber at 20 °C with 16 h light (80 µmol m^−2^s^−1^) and 60% relative humidity. Six days after germination, the seedlings were robust and shifted onto sterile filter papers soaked with half-strength Murashige and Skoog (MS) medium (Sigma-Aldrich, St. Louis, MO, USA) (supplemented with 0.5% sucrose and 5.4 µM 1-naphthylacetic acid), inserted into sterilized plant boxes and incubated in a growth chamber set to the conditions mentioned above until the seedlings were two weeks old.

### 4.2. Microbial Priming and Pathogen Infection

To analyse the effects of microbial priming in rapeseed defence activation, two-week-old rapeseed roots were primed with a mixed suspension of *Trichoderma harzianum* OMG16 (received from the strain collection of Anhalt University of Applied Sciences, Bernburg, Germany) plus *Bacillus velezensis* FZB42 syn. *Bacillus amyloliquefaciens* subsp. *plantarum* FZB42 (obtained from ABiTEP GmbH, Berlin, Germany). Microbial growth conditions and inoculation procedures were carried out according to the protocol mentioned in Hafiz et al. (2022) [53]. Briefly, roots of rapeseed seedlings were dipped in a combined inocula of OMG16 (2000 spores per mL) plus FZB42 (OD_600_ = 0.275, corresponding to approximately 2.2 × 10^8^ cells per mL) or in deionized water (control). Two days after priming, the same roots were infected with *Verticillium longisporum* 43 (provided by NPZ Innovation GmbH, Hohenlieth, Germany) by dipping into Vl43 spore suspension (10^6^ spores per mL) for 30 min or into deionized water (control). Roots of non-primed seedlings were also infected with Vl43 as positive controls. For each treatment, there were five biological replicates and three plants were pooled to obtain one biological replicate.

### 4.3. Microscopic Analysis of Vl43-Infected Rapeseed Leaf Tissue

Roots of two-week-old rapeseed seedlings were infected with Vl43 according to the procedure mentioned above. Eight days after infection, cotyledons were separated from the stem using a sharp scalpel. Prior to staining, leaves were thoroughly washed with sterile water. Then, the leaves were dipped in blue staining solution (25% acetic acid and 10% royal blue ink from Rohrer and Klingner Leipzig-Co., Zella-Mehlis, Germany) for 1 min. Stained leaves were briefly washed with sterile deionized water and examined under a Zeiss Axio Observer 7 inverse microscope (Carl Zeiss, Jena, Germany).

### 4.4. Absolute Quantification of Vl43 DNA in Root Samples via qPCR

Roots were harvested from both Vl43-infected primed and non-primed plants. Approximately 50 mg of root segments were weighed to isolate total DNA with the DNeasy plant mini kit (Qiagen, Venlo, The Netherlands) according to the manufacturer instructions. Then, the concentrations of the isolated DNA were measured with Qubit 3.0 fluorometer using the Qubit dsDNA assay kit (Thermo Fisher Scientific, Waltham, MA, USA). A qPCR reaction of 20 μL total volume contained 15 ng of isolated DNA from infected roots, 10 μL of PowerUp SYBR Green master mix (Applied Biosystems, Waltham, MA, USA) and the *V. longisporum*-specific primer pairs OLG70 and OLG71 [7]. The following qPCR program was used in a QuantStudio 5 real-time PCR system (Applied Biosystems, Waltham, MA, USA): 50 °C for 2 min, 95 °C for 2 min, and 36 cycles of amplification (denaturation, 95 °C for 20 s; annealing, 59 °C for 30 s and elongation, 72 °C for 30 s). A melting curve was included with initial denaturation at 95 °C for 15 s, annealing at 60 °C for 1 min, and finally a linear increase in the temperature from 60 to 95 °C with a rate of 0.1 °C s^−1^. All primer sequences used in this study are listed in Appendix A. The precise annealing temperature and efficiency of each primer pair was determined by gradient PCR and applied in qPCR (Appendix A).

### 4.5. Gene Expression Studies with Total RNA Extraction, cDNA Synthesis and qRT-PCR

To analyse the tissue-specific priming effects, root, stem and leaf tissues were separated with a sharp scalpel and immediately transferred to −80 °C at 1 day and 2 days after OMG16 and FZB42 priming. Moreover, the same tissue types were also collected and immediately stored in −80 °C at 1 day, 2 days, 3 days and 5 days after Vl43 infection. Approximately 60 mg of tissues were weighed to isolate total RNA by using the RNeasy plant mini kit (Qiagen, Venlo, The Netherlands), according to the manufacturer’s instructions. Then, 1.2 µg of total RNA was taken to synthesize cDNA by using the SuperScript IV first-strand synthesis system (Thermo Fisher Scientific, Waltham, MA, USA). The concentrations of isolated cDNAs were quantified with a Qubit 3.0 fluorometer, using the Qubit ssDNA assay kit (Thermo Fisher Scientific, Waltham, MA, USA). The expression of rapeseed defence-related genes was analysed using qRT-PCR performed in a QuantStudio 5 real-time PCR system (Applied Biosystems, Waltham, MA, USA). qRT-PCR reactions were prepared in quadruplicate, each in 20 µL of total volume containing 15 ng of cDNA, 10 µL of PowerUp SYBR Green master mix (Applied Biosystems, Waltham, MA, USA) and 5 pmol each of forward and reverse primer. The cycling instructions for qRT-PCR included UGD activation at 50 °C for 2 min, Dual-Lock^TM^ DNA polymerase at 95 °C for 2 min and 40 cycles of amplification (denaturation, 95 °C for 1 s; annealing and elongation, 60 °C for 30 s). A melting curve was recorded after amplification with initial denaturation at 95 °C for 15 s, annealing at 60 °C for 1 min and finally a linear increase in the temperature from 60 °C to 95 °C with a rate of 0.1 °C s^−1^. For normalization, *BnaUbiquitin11*, *BnaActin* and *BnaTubulin* were selected to serve as endogenous controls. To calculate the relative transcript abundances of genes, the efficiency-corrected Cq of three endogenous controls was averaged. ∆Cq was calculated as follows: ∆Cq = Cq reference genes—Cq target gene [75,76,77].

### 4.6. Determination of Phytohormones Using UHPLC-MS Analysis

After microbial inoculation onto rapeseed roots, as mentioned above, roots and shoots were harvested by cutting with a sharp scalpel and tissues were frozen separately into liquid nitrogen. There were four biological replicates, and three plants were pooled for each biological replicate. Phytohormone contents of rapeseed shoots and roots were determined with UHPLC-MS analysis according to the procedure mentioned in Moradtalab et al. (2018), with slight modifications [78]. Briefly, 1 g of frozen tissue samples (shoots, roots) was ground to fine powder with liquid N_2_ and extracted with 80% methanol (5 mL). Afterwards, the ground tissues were further homogenized by ultrasonication (Micra D-9 homogenizer, Art, Müllheim, Germany) for 1 min and 15 s at 10,000 rpm, and then 2 mL methanol extracts were added to microtubes and centrifuged at 14,000× *g* for 10 min. Later, the supernatant (400 µL) was mixed with same amount of ultra-pure water and again centrifuged at 14,000× *g* for 5 min. The supernatant was cleaned with membrane filtration (Chromafil R O20/15 MS) and transferred to HPLC vials. UHPLC-MS analysis was conducted in a Velos LTQ System (Thermo Fisher Scientific, Waltham, MA, USA) fitted with a Synergi Polar column, 4 µL, 150 × 3.0 mm (Phenomenex, Torrance, CA, USA), for the measurement of JA and SA phytohormones. The injection volume was set to 3 µL. The flow rate was adjusted to 0.5 mL min^−1^ for gradient elution with mobile phase A (water/acetonitrile = 95/5 (*v*/*v*)) and mobile phase B (pure acetonitrile), and a gradient profile of 0–1 min, 95% A, 5% B, 11–13 min, 10% A, 90% B, 13.1 min, 95% A, 5% B, 16 min 95% A and 5% B. All standard reference substances were purchased from Sigma Aldrich (St. Louis, MO, USA).

### 4.7. Statistical Analysis

In this study, statistical analysis was performed using R software, version 4.0.2 [79]. The significance of rapeseed defence gene expression after Vl43 pathogen challenge was calculated using Dunnett’s test with multivariate testing (mvt) adjustment at *p* < 0.05 compared to the untreated control plants. The significance of the relative expression of genes between Vl43-challenged primed and non-primed treatments was calculated using an unpaired *t*-test at *p* < 0.05. The significance of endogenous plant hormone concentrations was calculated using Dunnett’s test with mvt adjustment at *p* < 0.05 compared to the untreated controls.

## 5. Conclusions

Our studies conducted so far demonstrated that priming rapeseed roots with *T. harzianum* OMG16 and *B. velezensis* FZB42 activated the plant defence system and systematically transduced that defence signals went from the root to the upper part of the plant body through the activation of a set of phytohormones. In Vl43-infected non-primed plants, SA biosynthesis and signalling was downregulated in roots compared to the untreated controls. Evidently, weaker and slower JA/ET signals were transduced from the roots to the leaves in Vl43-infected non-primed plants compared to the primed plants. On the other hand, in primed plants, SA biosynthesis and signalling were upregulated in roots after Vl43 infection compared to the non-primed plants. Activated SA simultaneously triggered both JA and ET biosynthetic pathways. In stems of primed plants, both JA and ET acted antagonistically with the SA biosynthesis pathway. Similarly, in leaves, elevated levels of JA suppressed SA production. In primed plants, the expression of the JA and ET marker genes was stronger and faster after Vl43 infection than in the non-primed plants. Here, remarkably intense SA, JA and ET signals were systematically transduced from the root to the shoot of the primed plants. A conceptional model summarizing the events of signal transduction involved in resistance induction against Vl43 in oilseed rape seedlings is presented in Figure 7. Our results suggest an alternative approach to develop enhanced rapeseed tolerance against *Verticillium* root infection. Such an induction of defence mechanisms upon beneficial microbial treatments could also be transferred to field conditions to further investigate the control of *Verticillium* stem stripping disease in rapeseed.

## Figures and Tables

**Figure 1 ijms-24-10489-f001:**
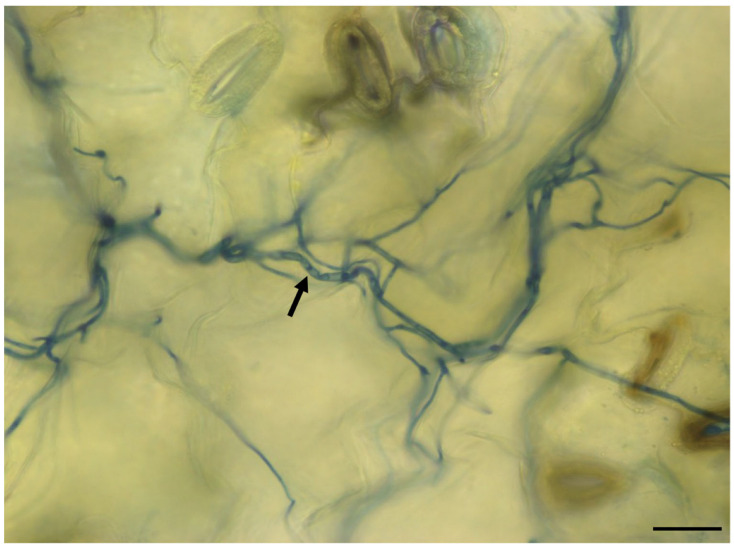
Endophytic colonization of rapeseed leaf with *Verticillium* hyphae observed after root infection. Roots of two-week-old plants were infected with Vl43 spores and leaves were stained with royal blue ink 8 days after infection. Arrows point to hyphae stained with royal blue ink in a leaf. Scale bar, 20 μm.

**Figure 2 ijms-24-10489-f002:**
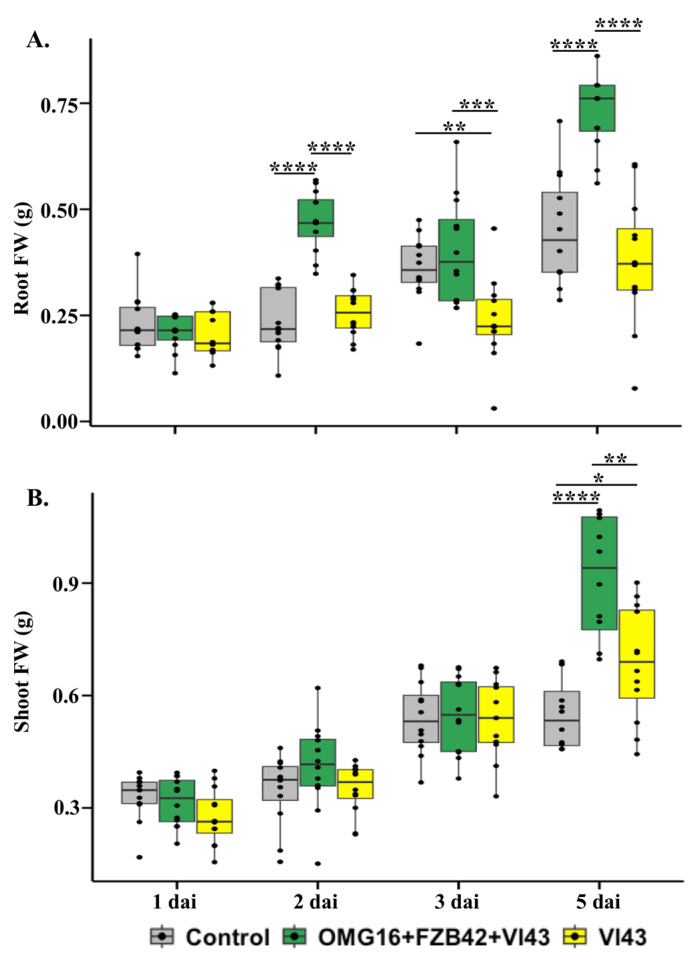
Effects of OMG16 and FZB42 priming and Vl43 infection on root and shoot development. Two-week-old rapeseed roots were primed with OMG16 plus FZB42 inocula. Two days later, the roots of the primed plants were infected with Vl43 fungal pathogen. As positive controls, roots of non-primed seedlings were also challenged with Vl43. Box and whisker plots show results of twelve biological replicates, where statistical significance was calculated using Dunnett’s test with multivariate testing (mvt) adjustment at *p* < 0.05 compared to the untreated control plants. Root (**A**) and shoot (**B**) fresh weight (FW) between OMG16 plus FZB42 plus Vl43 and single Vl43 treatments were calculated with Wilcoxon rank-sum tests at *p* < 0.05. * *p* < 0.05; ** *p* < 0.01; *** *p* < 0.001; **** *p* < 0.0001. OMG16, *T. harzianum* OMG16; FZB42, *B. velezensis* FZB42; Vl43, *V. longisporum* 43; dai, days after infection.

**Figure 3 ijms-24-10489-f003:**
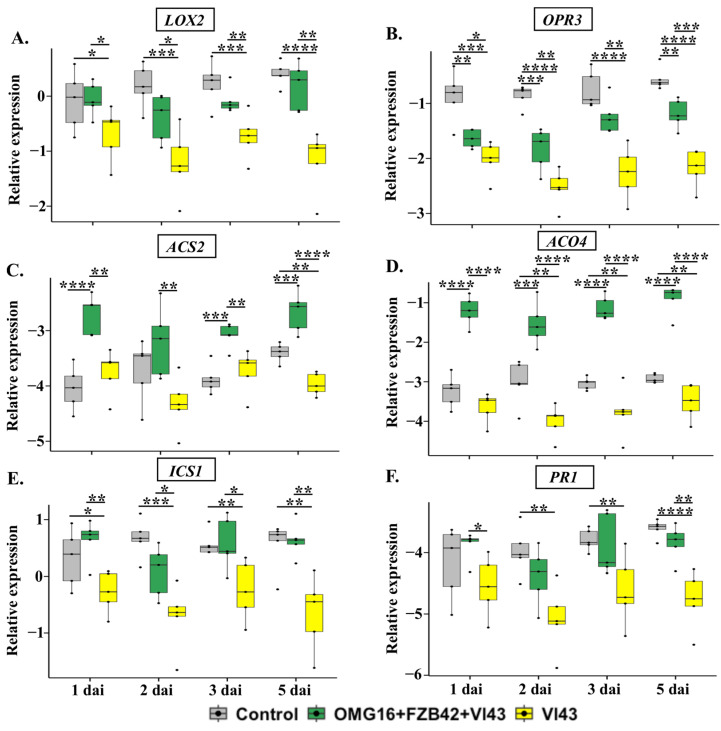
Relative transcript abundances of defence-related rapeseed genes *LOX2* (**A**), *OPR3* (**B**), *ACS2* (**C**), *ACO4* (**D**), *ICS1* (**E**) and *PR1* (**F**) in roots upon Vl43 infection in primed (OMG16 + FZB42) and non-primed plants compared to untreated plants. Priming was performed by inoculating the roots of two-week-old rapeseed seedlings with OMG16 plus FZB42. Two days later, the roots of the primed plants were challenged with pathogenic Vl43. To assess local responses, the relative transcript abundance of the rapeseed defence genes involved in JA, ET and SA biosynthesis and signalling pathways were analysed using qRT-PCR. As endogenous reference genes, *BnaActin*, *BnaTubulin* and *BnaUbiquitin11* were applied for normalization. ΔCq values were calculated from five biological replicates with quadruplicate qRT-PCRs and represented in box and whisker plots. Dunnett’s test was applied with mvt adjustment at *p* < 0.05 to enumerate the significance of changes in gene expression compared to the untreated control plants. An unpaired *t*-test was also applied at *p* < 0.05 to calculate the significances of differences in relative gene expression between OMG16 plus FZB42 plus Vl43 and single Vl43 treatments. * *p* < 0.05; ** *p* < 0.01; *** *p* < 0.001; **** *p* < 0.0001. OMG16, *T. harzianum* OMG16; FZB42, *B. velezensis* FZB42; Vl43, *V. longisporum* 43; dai, days after infection.

**Figure 4 ijms-24-10489-f004:**
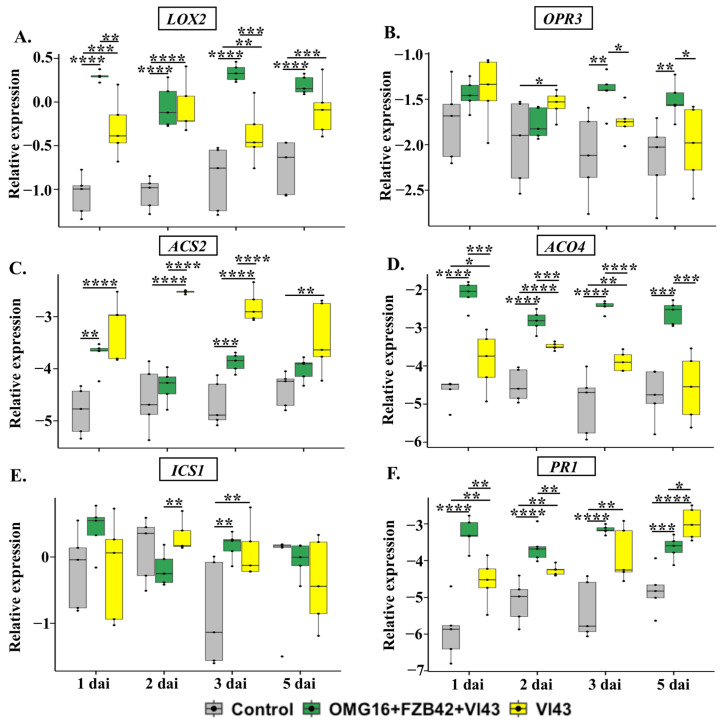
Relative expression of rapeseed defence genes *LOX2* (**A**), *OPR3* (**B**), *ACS2* (**C**), *ACO4* (**D**), *ICS1* (**E**) and *PR1* (**F**) in stems upon OMG16 plus FZB42 plus Vl43 and Vl43 inoculation alone. Priming was performed by inoculating the roots of two-week-old rapeseed seedlings with OMG16 plus FZB42. Two days later, the roots of the primed plants were challenged with pathogenic Vl43. To assess systemic responses, the relative transcript abundance of the rapeseed defence genes involved in JA, ET and SA biosynthesis and signalling pathways were analysed with qRT-PCR in stems of the same plants tested in Figure 3. As endogenous reference genes, *BnaActin*, *BnaTubulin* and *BnaUbiquitin11* were applied for normalization. ΔCq values were calculated from five biological replicates with quadruplicate qRT-PCRs and represented in box and whisker plots. Dunnett’s test was applied with mvt adjustment at *p* < 0.05 to enumerate the significance of changes in gene expression compared to the untreated control plants. An unpaired *t*-test was applied at *p* < 0.05 to calculate the significances of differences in relative gene expression between OMG16 plus FZB42 plus Vl43 and single Vl43 treatments. * *p* < 0.05; ** *p* < 0.01; *** *p* < 0.001; **** *p* < 0.0001. OMG16, *T. harzianum* OMG16; FZB42, *B. velezensis* FZB42; Vl43, *V. longisporum* 43; dai, days after infection.

**Figure 5 ijms-24-10489-f005:**
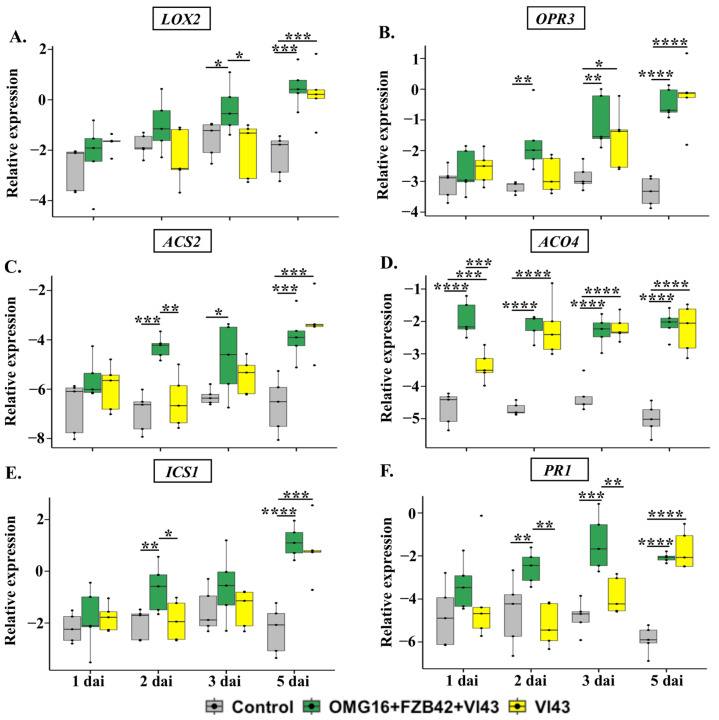
Relative expressions of rapeseed defence genes *LOX2* (**A**), *OPR3* (**B**), *ACS2* (**C**), *ACO4* (**D**), *ICS1* (**E**) and *PR1* (**F**) in leaves upon OMG16 plus FZB42 plus Vl43 and Vl43 inoculation alone. Priming was performed by inoculating the roots of two-week-old rapeseed seedlings with OMG16 plus FZB42. Two days later, the roots of the primed plants were challenged with pathogenic Vl43. To assess systemic responses, the relative transcript abundance of the rapeseed defence genes involved in JA, ET and SA biosynthesis and signalling pathways were analysed using qRT-PCR in leaves of the same plants tested in Figure 3. As endogenous reference genes, *BnaActin*, *BnaTubulin* and *BnaUbiquitin11* were applied for normalization. ΔCq values were calculated from five biological replicates with quadruplicate qRT-PCRs and represented in box and whisker plots. Dunnett’s test was applied with mvt adjustment at *p* < 0.05 to enumerate the significance of changes in gene expression compared to the untreated control plants. An unpaired *t*-test was applied at *p* < 0.05 to calculate the significances of differences in relative gene expression between OMG16 plus FZB42 plus Vl43 and single Vl43 treatments. * *p* < 0.05; ** *p* < 0.01; *** *p* < 0.001; **** *p* < 0.0001. OMG16, *T. harzianum* OMG16; FZB42, *B. velezensis* FZB42; Vl43, *V. longisporum* 43; dai, days after infection.

**Figure 6 ijms-24-10489-f006:**
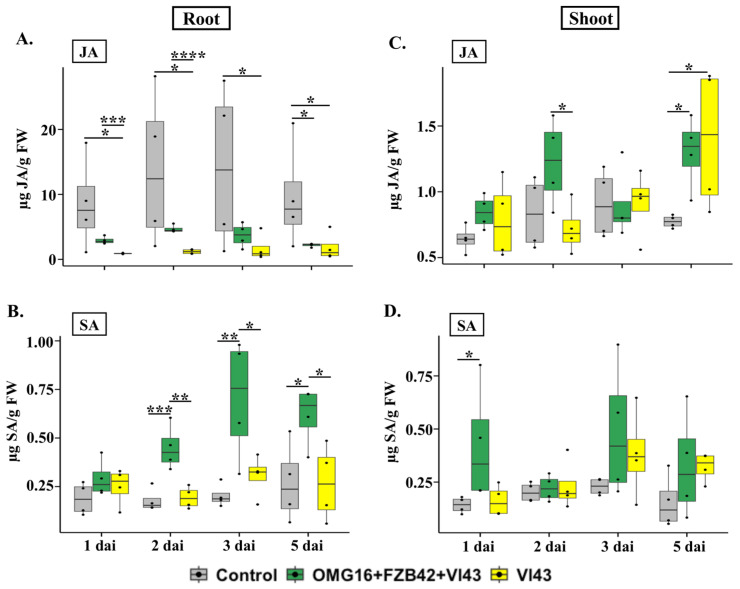
Endogenous contents of JA (**A**,**C**) and SA (**B**,**D**) in root and shoot tissues of rapeseed plants at different time points upon OMG16 plus FZB42 plus Vl43 and Vl43 inoculation alone. Priming was performed by inoculating the roots of two-week-old rapeseed seedlings with OMG16 plus FZB42. Two days later, the roots of the primed plants were challenged with pathogenic Vl43. To assess the phytohormone content, roots and shoots (both leaves and stems) of the same plants were separately analysed using UHPLC-MS. Box and whisker plots show the phytohormone levels of five biological replicates where significance was calculated by Dunnett’s test with mvt adjustment at *p* < 0.05 compared to the untreated control plants. Phytohormone content between OMG16 plus FZB42 plus Vl43 and single Vl43 treatments was calculated by unpaired *t*-test at *p* < 0.05. * *p* < 0.05; ** *p* < 0.01; *** *p* < 0.001; **** *p* < 0.0001. OMG16, *T. harzianum* OMG16; FZB42, *B. velezensis* FZB42; Vl43, *V. longisporum* 43; dai, days after infection.

**Figure 7 ijms-24-10489-f007:**
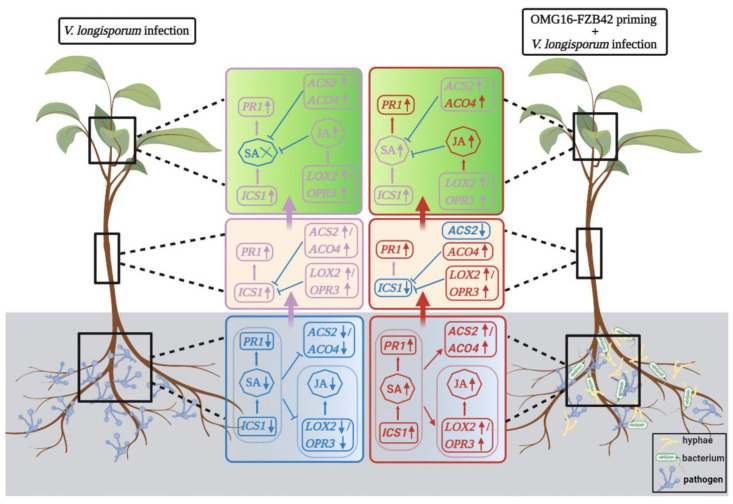
Schematic representation of molecular mechanisms involved in rapeseed defence signal transduction from root to stem to leaf tissues through crosstalk among the SA, ET and JA hormonal pathways in response to *Verticillium*-rapeseed interaction and *Trichoderma*-*Bacillus*-rapeseed priming, followed by *Verticillium* infection. The red colour represents faster and stronger responses, while purple represents slower and weaker responses. Moreover, the blue colour represents down-regulation or suppression. Arrows indicate positive regulation (activation) and blunt-ended lines indicate negative regulation (repression), while upward arrows (↑) indicate upregulation and downward arrows (↓) indicate downregulation.

**Table 1 ijms-24-10489-t001:** Quantification of Vl43 DNA in root tissue of Vl43-infected primed and non-primed plants with qPCR.

Treatment	Vl43 Infection	OMG16-FZB42 Priming + Vl43 Infection
Days after Infection (dai)	ng Vl43 DNA from 15 ng Root DNA	ng Vl43 DNA from 15 ng Root DNA
1	40.8 (±4.8)	2.1 (±0.3) ****
2	111.4 (±21.7)	3.5 (±1.0) ***
3	229.5 (±33.8)	1.4 (±0.7) ****
5	271.7 (±15.8)	15.2 (±9.5) ****

Values are means of the amount of Vl43 DNA in roots, ±standard error (*** *p* < 0.001; **** *p* < 0.0001).

## Data Availability

Data will be made available upon request.

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
