# Peer review of "Tissue-Specific Hormone Signalling and Defence Gene Induction in an In Vitro Assembly of the Rapeseed Verticillium Pathosystem"

_ijms, 2023, doi:10.3390/ijms241310489_

Round 1
Reviewer 1 Report (Previous Reviewer 2)
I see that you have changed the text a little and improved the English grammar.
You corrected the places where I wrote to you last time. I thank you. It is good now.
I have only a few corrections:
Line 208: Change the word pants to plants.
Line 320: Put the year 2007 in regular parentheses. You do that in line 421.
Line 510: The same reasoning as in the previous comment.
Lines 518 and 519: I think you should use one title since you do not have two subtitles, you have one. I suggest you reword the title.
Lines 542 and 543: Same situation as in the previous comment. I suggest you to reword the title or rename it somehow, but you decide.
Line 573: I think you should put regular brackets instead of square brackets.
Author Response
Reviewer 1
I see that you have changed the text a little and improved the English grammar.
You corrected the places where I wrote to you last time. I thank you. It is good now.
I have only a few corrections:
Line 208: Change the word pants to plants.
The spelling “plants” has been corrected.
Line 320: Put the year 2007 in regular parentheses. You do that in line 421.
Year 2007 has been put in regular parentheses.
Line 510: The same reasoning as in the previous comment.
Year 2022 has been put in regular parentheses.
Lines 518 and 519: I think you should use one title since you do not have two subtitles, you have one. I suggest you reword the title.
We have removed the subtitle and rearranged the sentence.
Lines 542 and 543: Same situation as in the previous comment. I suggest you to reword the title or rename it somehow, but you decide.
We have removed the subtitle and rearranged the sentence.
Line 573: I think you should put regular brackets instead of.
Regular brackets were included, and square brackets were removed.
Reviewer 2 Report (New Reviewer)
ijms-2384199-peer-review-v1
The manuscript ‘Tissue Specific Hormone Signalling and Defence Gene Induction in an In-vitro Assembly of the Verticillium-Rapeseed Patho-system’ by Hazif et al. is well-planned, well-executed, and well-written. However, I have a few suggestions which could improve the manuscript.
Line 3-4: Please remove the ‘-‘ between ‘In- vitro’ and ‘patho-system’ in the title. Please revise as ‘In vitro’ and ‘pathosystem’.
‘Rapeseed Verticillium’ is more preferred than ‘Verticillium-Rapeseed’, please revise.
Line 19: Please check the underlined and colored words throughout the text.
Plant morphology is irrelevant in the keywords, authors may remove it.
Line 112-113: Please mention days after inoculation/infection as ‘DAI’
Line 115: highly enhanced…..please quantify
Line 117-121: Please quantify.
Table 1: DNA nicking assay would be more appropriate than DNA quantification.
Please mention the values ‘2.1****’ (±0.3)’ as ‘2.1(±0.3)****’
Please make the “Values are means of the amount of Vl43 DNA in roots; ±standard error. t-test: *** P <0.001; **** P 146 ><0.0001” as “Values are means of the amount of Vl43 DNA in roots ± standard error (*** P <0.001; **** P 146 ><0.0001)”.
Line 493-494: Surface sterilization of rapeseed in 3% sodium hypochlorite solution for 15 min is too high. Please cross-check.
Line 500-502: Why the 6-day-old germinated seedlings were transferred to MS-NAA medium and inserted into sterilized plant boxes? Please justify. Please simplify the process.
Line 531: (Qiagen)…Please mention the city and country for all the equipment and consumables used in the study.
Line 544: Whether, Vl43 infection quantification in root, stem and leaf tissues has been done to justify the tissue-specific priming effects?
Line 589: Please retain the City, State, and Country for the chemicals used in the study.
Please reduce the references to 45-50. Authors may use the link of the software used in the study followed by the date of assess instead of citing a reference [64].
Authors may consider discussing the following citations on tissue-specific gene actions;
https://doi.org/10.3389/fpls.2020.584997
https://doi.org/10.1371/journal.pone.0246971
References may be arranged as per the journal pattern. Please cross-check the references cited with the list and text.
Although the overall grammar score is satisfactory, the authors are requested to polish the language, grammar, and punctuation again while doing the minor revision.
The manuscript may be accepted with minor corrections.
Good luck with the revision.
Minor English is required
Author Response
Reviewer 2
The manuscript ‘Tissue Specific Hormone Signalling and Defence Gene Induction in an In-vitroAssembly of the Verticillium-Rapeseed Patho-system’ by Hazif et al. is well-planned, well-executed, and well-written. However, I have a few suggestions which could improve the manuscript.
Line 3-4: Please remove the ‘-‘ between ‘In- vitro’ and ‘patho-system’ in the title. Please revise as ‘In vitro’ and ‘pathosystem’.
‘Rapeseed Verticillium’ is more preferred than ‘Verticillium-Rapeseed’, please revise.
The “-“ between ‘In- vitro’ and ‘patho-system’ was removed in the title and rewritten as ‘In vitro’ and ‘pathosystem’.
Line 19: Please check the underlined and colored words throughout the text.
The underlined and colored words were checked throughout the text and changed accordingly.
Plant morphology is irrelevant in the keywords, authors may remove it.
“Plant morphology” was removed from the key word.
Line 112-113: Please mention days after inoculation/infection as ‘DAI’
“Days after infection (dai)” was included in the text.
Line 115: highly enhanced…..please quantify
We have quantified the elevated amount of root fresh weight in Vl43 infected primed plants compared to the non-primed plants and mentioned in the discussion part in section 3.2.
Line 117-121: Please quantify.
The increased amount of shoot fresh weight was also quantified and discussed in detail in section 3.2.
Table 1: DNA nicking assay would be more appropriate than DNA quantification.
The authors would like to thank the reviewer for this suggestion. We have slightly modified the title of Table 1. Now “detection” was substituted by “quantification”.
Please mention the values ‘2.1****’ (±0.3)’ as ‘2.1(±0.3)****’
The asterisks were rearranged as suggested.
Please make the “Values are means of the amount of Vl43 DNA in roots; ±standard error. t-test: *** P <0.001; **** P 146 ><0.0001” as “Values are means of the amount of Vl43 DNA in roots ± standard error (*** P <0.001; **** P 146 ><0.0001)”.
The above-mentioned sentence has been changed as suggested.
Line 493-494: Surface sterilization of rapeseed in 3% sodium hypochlorite solution for 15 min is too high. Please cross-check.
This statement is correct. Despite this strong treatment, all seeds were viable and with high germination rate.
Line 500-502: Why the 6-day-old germinated seedlings were transferred to MS-NAA medium and inserted into sterilized plant boxes? Please justify. Please simplify the process.
The sterilized seeds were germinated together on a glass Petri dish. 6 days after germination, each of the young seedlings was individually transferred to single sterile plant boxes containing growth medium (MS-NAA) for further treatment with beneficials and pathogen. This has been clarified in the text.
Line 531: (Qiagen)…Please mention the city and country for all the equipment and consumables used in the study.
The cities and countries were mentioned for all the equipment and consumables used in the study.
Line 544: Whether, Vl43 infection quantification in root, stem and leaf tissues has been done to justify the tissue-specific priming effects?
The relative defence gene expression was indeed analysed in root, stem and leaf tissues to access the tissue-specific priming effects in rapeseed. This was clarified in the text.
Line 589: Please retain the City, State, and Country for the chemicals used in the study.
The cities, stated, and countries were mentioned for all the chemicals used in the study.
Please reduce the references to 45-50. Authors may use the link of the software used in the study followed by the date of assess instead of citing a reference [64].
The authors would prefer to cite the software as a reference and write the details in the reference list.
Authors may consider discussing the following citations on tissue-specific gene actions;
https://doi.org/10.3389/fpls.2020.584997
https://doi.org/10.1371/journal.pone.0246971
The authors thank the reviewer for this suggestion. Article Dhar et al. 2020 was added in the manuscripts. We feel that Devi et al., 2021 was not appropriate for citation because the plant regeneration from root explants was different from our approach.
References may be arranged as per the journal pattern. Please cross-check the references cited with the list and text.
References have been cross-checked with the list and the text.
Although the overall grammar score is satisfactory, the authors are requested to polish the language, grammar, and punctuation again while doing the minor revision.
Many thanks for this comment. We have improved language, grammar, and punctuation throughout the manuscript.
The manuscript may be accepted with minor corrections.
Good luck with the revision.
Reviewer 3 Report (New Reviewer)
The article is devoted to the actual topic of plant protection from pathogens with the help of beneficial microorganisms and the study of the priming mechanism using the Verticillium-Rapeseed Patho-system as an example. The results obtained by the authors are of scientific interest, but the article is written carelessly, the results are poorly presented. The manuscript needs extensive revision.
General comments.
1. The review of the literature is not thorough so the reader is not given an adequate background about the topic. Authors can improve the introduction part and discuss it.
In addition, outdated references are used in the introduction. In the last 3 years, a lot of articles have been published on the induction of ISR by bacteria and endophytic fungi and on the mechanism of priming.
2. The Results section is sloppy. The authors should improve it. Figures 2, 3, 4, 5 and 6 need to be improved. The authors used similar colors for control and infection variants in these Figures. Colors should be contrasting.
3. The discussion section repeats the description of the results, there is not comparison with previously published results and current work. Only 12-14 sources were used in the discussion and all were published almost 10 years ago. The discussion should be rewritten using current literature.
Specific remarks:
Lines 113-114: In contrary, root weight was decreased on 3 dai (P < 0.01) in the Vl43 infected non-primed plants.
What is the weight loss compared to? And further throughout the text of the Results. Lines 118, 169, 207, 254.
Lines 169-172: Analysis of ET biosynthesis gene expression in the roots of primed plants revealed that the mRNA abundances of ACS2 and ACO4 were downregulated compared to the untreated controls. However, this effect was completely reverted by priming the roots with the selected OMG16 and FZB42 inoculants prior to Vl43 infection (Figure 3C, D).
These 2 sentences are misleading. Which plants are being referred to in the first sentence, infected or not? The sentences should be rewritten. Insert a reference to the Figure or Supplementary Materials.
Lines 207-209: Contrary, the expression of ACO4 was massively increased in primed Vl43 infected pants while it was only slightly enhanced or remained even at a level comparable to control plants (Figure 4D).
The same comment.
Lines 291-294: The content of SA increased on 2 dai (P <0.001), 3 dai (P <0.01) and 5 dai (P <0.05) in Vl43 infected primed plants. In contrast, there was no induction of SA in Vl43 infected non-primed plants compared to the untreated controls. The level of endogenous SA was enhanced in roots of Vl43 infected primed plants on 2 dai (P <0.01), 3 dai (P <0.05) and 5 dai (P <0.05) compared to the non-primed controls (Figure 6B).
These 3 sentences are misleading. Which plants are being referred to in the first sentence, infected or not? The sentences should be rewritten.
Lines 242-244: Gene name repetitions can be removed.
Lines 276-282: This paragraph should be transferred to the Methods section
Line 324: There is no morphology in the results. Authors should explain.
Lines 480-482: In a similar study, rhizobacterium P. alvei K165 triggered ISR to protect A. thaliana from V. dahliae infestation in SA jasmonate-dependent defence pathways by increasing PR1 expression in leaves [57].
This sentence is not clear and should be rewritten.
Lines 507-508, 513: Authors should indicate the sources of microorganisms.
Lines 519-527: Rewrite section 4.3, especially the last sentence, is not written in scientific language.
Author Response
Reviewer 3
The article is devoted to the actual topic of plant protection from pathogens with the help of beneficial microorganisms and the study of the priming mechanism using the Verticillium-Rapeseed Patho-system as an example. The results obtained by the authors are of scientific interest, but the article is written carelessly, the results are poorly presented. The manuscript needs extensive revision.
General comments.
- The review of the literature is not thorough so the reader is not given an adequate background about the topic. Authors can improve the introduction part and discuss it.
In addition, outdated references are used in the introduction. In the last 3 years, a lot of articles have been published on the induction of ISR by bacteria and endophytic fungi and on the mechanism of priming.
Many thanks for this comment. We have improved the introduction and included a number of important recent studies.
- The Results section is sloppy. The authors should improve it. Figures 2, 3, 4, 5 and 6 need to be improved. The authors used similar colors for control and infection variants in these Figures. Colors should be contrasting.
The authors agree that the colours used in Figures 2, 3, 4, 5 and 6 were similar. Therefore, contrasting colors were introduced in Figures 2, 3, 4, 5 and 6 in the revised manuscript.
- The discussion section repeats the description of the results, there is not comparison with previously published results and current work. Only 12-14 sources were used in the discussion and all were published almost 10 years ago. The discussion should be rewritten using current literature.
The author would like to thank the reviewer for this comment. We have improved the discussion part and included a number of important and relevant recent studies.
Specific remarks:
Lines 113-114: In contrary, root weight was decreased on 3 dai (P < 0.01) in the Vl43 infected non-primed plants.
What is the weight loss compared to? And further throughout the text of the Results. Lines 118, 169, 207, 254.
Response: In lines 113-114, root weight loss of Vl43 infected non-primed plants was compared to the untreated control plants. In the revised manuscript the sentence was modified. Following text was also modified according to the reviewer’s suggestion.
Lines 169-172: Analysis of ET biosynthesis gene expression in the roots of primed plants revealed that the mRNA abundances of ACS2 and ACO4 were downregulated compared to the untreated controls. However, this effect was completely reverted by priming the roots with the selected OMG16 and FZB42 inoculants prior to Vl43 infection (Figure 3C, D).
These 2 sentences are misleading. Which plants are being referred to in the first sentence, infected or not? The sentences should be rewritten. Insert a reference to the Figure or Supplementary Materials.
Response: In the first sentence, we referred to Vl43 infected non-primed plants. This sentence has been rewritten in the revised manuscript.
Lines 207-209: Contrary, the expression of ACO4 was massively increased in primed Vl43 infected pants while it was only slightly enhanced or remained even at a level comparable to control plants (Figure 4D).
The same comment.
Response: Lines 207-209 have been rewritten in the revised manuscript.
Lines 291-294: The content of SA increased on 2 dai (P <0.001), 3 dai (P <0.01) and 5 dai (P <0.05) in Vl43 infected primed plants. In contrast, there was no induction of SA in Vl43 infected non-primed plants compared to the untreated controls. The level of endogenous SA was enhanced in roots of Vl43 infected primed plants on 2 dai (P <0.01), 3 dai (P <0.05) and 5 dai (P <0.05) compared to the non-primed controls (Figure 6B).
These 3 sentences are misleading. Which plants are being referred to in the first sentence, infected or not? The sentences should be rewritten.
Response: In lines 291-294, the content of SA increased on 2 dai (P <0.001), 3 dai (P <0.01) and 5 dai (P <0.05) in Vl43 infected primed plants compared to the untreated controls. We have modified these sentences in the revised version.
Lines 242-244: Gene name repetitions can be removed.
Response: Gene names have been removed.
Lines 276-282: This paragraph should be transferred to the Methods section
Response: We agree that this paragraph is somehow methodological. However, it summarises the experimental setup briefly and introduces thereby the following sections. Since the Materials and Methods section is in IJMS at the end of the manuscript we think that a brief description of the experiments improves readability of the manuscript. Therefore, we would like to keep these sentences at the beginning of section 2.5.
Line 324: There is no morphology in the results. Authors should explain.
Response: Thank you for the comment. We erased “the morphology” because it is not appropriate in this context.
Lines 480-482: In a similar study, rhizobacterium P. alvei K165 triggered ISR to protect A. thaliana from V. dahliae infestation in SA jasmonate-dependent defence pathways by increasing PR1 expression in leaves [57].
This sentence is not clear and should be rewritten.
Response: Lines 480-482 have been rewritten.
Lines 507-508, 513: Authors should indicate the sources of microorganisms.
Response: The sources of microorganisms have been included.
Lines 519-527: Rewrite section 4.3, especially the last sentence, is not written in scientific language.
Response: We agree that some improvements needed to be done here. Therefore, we have carefully checked section 4.3 and made necessary changes.
Reviewer 4 Report (New Reviewer)
The topic is very interesting, experiments are well designed and performed, and it is well written, thus I recommend the manuscript for publication. The text has been already improved by a previous revision.
A previous work of authors (Hafiz et al 2022 Mol. Plant-Microbe Interact. 2022, 35, 380-392, doi:10.1094/mpmi-11-21-0274-r.) already report that T. harzianum OMG16 and B. velezensis FZB42 priming in rapeseed induce defense activation against V. longisporum 43 (Vl43) infection and that JA/ET signaling is involved because some related genes expressions are induced. The present manuscript is an in-depth analysis of these findings.
In the manuscript an experiment has been added after the revision(?) that analyzed the average amount of VL43 DNA by qPCR in primed and non-primed rapeseed infected roots to quantify priming effects on reducing root infection by Vl43 (chapters 2.3, 3.3 and 4.4). This experiment has already been reported in Hafiz et al. 2022.
More comments and suggestions are given in the file attached.

Author Response
Reviewer 4
The topic is very interesting, experiments are well designed and performed, and it is well written, thus I recommend the manuscript for publication. The text has been already improved by a previous revision.
A previous work of authors (Hafiz et al 2022 Mol. Plant-Microbe Interact. 2022, 35, 380-392, doi:10.1094/mpmi-11-21-0274-r.) already report that T. harzianum OMG16 and B. velezensis FZB42 priming in rapeseed induce defense activation against V. longisporum 43 (Vl43) infection and that JA/ET signaling is involved because some related genes expressions are induced. The present manuscript is an in-depth analysis of these findings.
In the manuscript an experiment has been added after the revision(?) that analyzed the average amount of VL43 DNA by qPCR in primed and non-primed rapeseed infected roots to quantify priming effects on reducing root infection by Vl43 (chapters 2.3, 3.3 and 4.4). This experiment has already been reported in Hafiz et al. 2022.
The authors would like to thank the reviewer for this detailed comment. We have added this data because the editor suggested to do so, and we think that these data are important since they show that the infection in the plant samples used in this study were successfully infected and that priming was highly efficient to reduce infection of the plants. However, we agree that we should mention that these results are in line with our previous finding. We have added a brief explanation in the revised manuscript.
More comments and suggestions are given in the file attached.
This manuscript is a resubmission of an earlier submission. The following is a list of the peer review reports and author responses from that submission.
Round 1
Reviewer 1 Report
The manuscript is an original paper that present results of the treatment of rapeseed plants with Trichoderma harzianum OMG16 and Bacillus velezensis FZB42 in order to induce defence activation against Verticillium longisporum infection.
In my opinion, the manuscript is very well work-out, the research methodology is correct and raises no objections. This article made a positive impression on me. While reading the manuscript, I have some minor comments which I have included below:
· Line 133-138: I think it is no need to repeat description of methodology that there is in the Material and methods chapter · In the Discussion: the proportions between the results obtained in the research described in the publication and the literature data are disturbed. There is no need to repeat the description of the results, the authors did not pay enough attention to the discussion of the results. Maybe a good solution would be to combine Result with Discussion in one chapter. · Line 451: please add explanation of OMG16 and FZB42Author Response
Response to Reviewer 1 Comments
Dear reviewer 1,
On behalf of the co-authors, we would like to thank you for carefully evaluating our manuscript and giving your valuable suggestions intended to improve the manuscript. We have revised our article, went through all the comments, and corrected accordingly. The revised manuscript has been uploaded.
The detailed responses to the comments of reviewer 1 are discussed as follows:
The manuscript is an original paper that present results of the treatment of rapeseed plants withTrichoderma harzianum OMG16 and Bacillus velezensis FZB42 in order to induce defence activation against Verticillium longisporum infection.
In my opinion, the manuscript is very well work-out, the research methodology is correct and raises no objections. This article made a positive impression on me. While reading the manuscript, I have some minor comments which I have included below:
- Line 133-138: I think it is no need to repeat description of methodology that there is in the Material and methods chapter ·
Response: The authors completely agree with the reviewer. Therefore lines 133-138 have been removed.
In the Discussion: the proportions between the results obtained in the research described in the publication and the literature data are disturbed. There is no need to repeat the description of the results, the authors did not pay enough attention to the discussion of the results. Maybe a good solution would be to combine Result with Discussion in one chapter. ·
Response: The authors would like to thank the reviewer for this comment. The discussion part was detailed in such a way that it would be easily accessible and understandable to the readers. Therefore, a short description of the main outcome and discussion in view of the relevant literatures were mandatory.
Line 451: please add explanation of OMG16 and FZB42
Response: A brief description of OMG16 and FZB42 has been added in line 451.
Reviewer 2 Report
The Introduction section is written well done. It is pretty clear written, the authors explained a lot of things which is necessary to understand the infection cycle of Verticillium longisporum, defence priming and what happens with JA, ET i SA phytohormone contents after infection with V. longisporum in primed and non-primed rapeseed plants.
The methodology described in the section Materials and Methods is well explained. I only think that you should have had more plants per biological replicate.
The results are explained well. It has a lot of results and these graphs are a little bit confused and it is difficult to understand quickly, but Figure 7 is very nice and explained everything.
I have a few corrections:
Line 181: S3 or S4?
Line 224: S4 or S5?
Line 314, 320, 321: Please check the numeration of figures in Supplementary material and make corrections in the text.
Line 343: Put a full stop at the end of the sentence.
Line 449, 500, 519, 521, 522, 528: Please check how you should write, µl or µL.
Line 517: Put the year in parentheses.
Line 563: Please put Verticillium italic.
Author Response
Response to Reviewer 2 Comments
Dear reviewer 2,
We would like to thank you for intensively reading our manuscript, reviewing it and giving your valuable suggestions which help us to improve the manuscript to a better scientific level. We very much appreciate the suggested modifications. We have revised our article, went through all of your comments and made the changes accordingly. The revised manuscript has been uploaded.
The detailed responses to the reviewers’ comments are discussed as follows:
The Introduction section is written well done. It is pretty clear written, the authors explained a lot of things which is necessary to understand the infection cycle of Verticillium longisporum, defence priming and what happens with JA, ET i SA phytohormone contents after infection with V. longisporum in primed and non-primed rapeseed plants.
The methodology described in the section Materials and Methods is well explained. I only think that you should have had more plants per biological replicate.
Response: There were 5 biological replicates for each treatment and 3 plants were pooled to obtain 1 biological replicate. Therefore, each data point represents the effects of 15 plants combined. Adding more plants for each treatment would exceed our limit for experimental set-up.
The results are explained well. It has a lot of results and these graphs are a little bit confused and it is difficult to understand quickly, but Figure 7 is very nice and explained everything.
I have a few corrections:
Line 181: S3 or S4?
Response: In line 181, it would be S4. In the revised manuscript, it has been corrected.
Line 224: S4 or S5?
Response: In line 224, it would be S5. In the revised manuscript, it has been corrected.
Line 314, 320, 321: Please check the numeration of figures in Supplementary material and make corrections in the text.
Response: The numeration of figures in Supplementary Materials has been checked in lines 314, 320, 321 and necessary corrections were made.
Line 343: Put a full stop at the end of the sentence.
Response: A full stop has been added at the end of the sentence.
Line 449, 500, 519, 521, 522, 528: Please check how you should write, µl or µL.
Response: Now µL was written everywhere to maintain uniformity.
Line 517: Put the year in parentheses.
Response: Parentheses have been added in the year. Now line 518.
Line 563: Please put Verticillium italic.
Response: In line 563,“Verticillium” is now written in Italic. Now line 566.